# Toughness Variations among Natural Casings: An Exploration on Their Biochemical and Histological Characteristics

**DOI:** 10.3390/foods11233815

**Published:** 2022-11-26

**Authors:** Wenjun Liu, Xing Chen, Satomi Tsutsuura, Tadayuki Nishiumi

**Affiliations:** 1Graduate School of Science and Technology, Niigata University, Niigata 950-2181, Japan; 2State Key Laboratory of Food Science and Technology, School of Food Science and Technology, Jiangnan University, Wuxi 214122, China

**Keywords:** hog and sheep casings, toughness, biochemical characteristic, histological characteristic, animal age

## Abstract

We investigated the mechanical, biochemical, and histological properties of hog and sheep casings produced in different countries to elucidate the responsible factors for the toughness quality of natural casings. The toughness and collagen characteristics of sheep and lamb casings were also investigated to elucidate the effect of animal slaughter age on the relationships between connective tissue and the mechanical properties of natural casings. The results showed that the main component of hog and sheep casings was collagen with many layers of sheets. The contents of collagen, elastin, and proteoglycan in hog and sheep casings were similar. The toughest Chinese casings (*p* < 0.01) possessed a significantly lower heat solubility of collagen (*p* < 0.01), and a different size and arrangement of collagen fibers. Sheep casings were significantly tougher than lamb casings (*p* < 0.01). Compared with lamb casings, sheep casings had a significantly low heat-labile collagen content, a low heat solubility of collagen, a large size of collagen fibers, and a high pyridinoline concentration (*p* < 0.01). Therefore, the high thermal and structural stability of collagen in aged animals may contribute to the enhanced mechanical properties of casings.

## 1. Introduction

Sausage casing, also known as sausage skin or simply casing, is the material that encloses the filling of a sausage. Nowadays, many types of sausage casings are used, including natural, manufactured collagen, cellulose, and plastic casings, as well as co-extruded casings made of collagen and alginate, and alginate-collagen hybrid casings [1]. Among them, natural casings are considered as the golden standard in sausage production owing to their desirable tenderness and high permeability to both moisture and smoke [2]. Natural casings are usually produced from the intestines of pigs, sheep, goats, and cattle (and horses, in certain cases). Among them, hog and sheep casings are mainly used in the sausage industry. The part of the small intestine often used in the sausage industry has a structure of tunica mucosa, submucosa, tunica muscularis, and tunica serosa from the inside. In hog and sheep casings, their submucosa, which is the remaining layer of the intestine after processing, forms the natural sausage casing [3].

The barrier properties and mechanical strength, which are two essential casing physical properties, strongly affect consumers’ perception of bite/snap and flavor [4]. There are still many quality defects in natural casings for sausage manufacturers, and their physical properties are required to be improved to meet the needs of consumers. Some hog and sheep casings with large diameters show high toughness. Therefore, when eating sausages, only the natural casing part remains in the mouth. There are even some hog sausage casings that need to be removed before eating. To suit consumers’ taste for soft sausages, the study performed by Sakata, Segawa, Morita, and Nagata [5] proposes the tenderization of hog casings with lactic acid and pepsin. On the other hand, excessive tenderization of natural casings is not acceptable. The strength and elasticity of a casing are also crucial during the processing of the sausage, as the casing must be strong enough to hold the meat but also able to expand during stuffing and cooking [6,7]. It is necessary to improve the quality of natural casings to an appropriate toughness by controlling its mechanical properties. However, little research has been performed on the mechanical, chemical, and histological properties and their relationships in natural casings.

Rupturing during stuffing is caused by gliding defects in natural casings, which is another problem for the industrial production of sausages. Houben, Bakker, and Keizer [8] reported that hog and sheep casings treated with trisodium phosphate facilitated the gliding of the casings over the test pipes and had a lower shear force (non-significant) than the control casings after being used as skins for cooked and smoked sausages. Hog casing modified with surfactant solutions combined with lactic acid has the best resistance to burst pressure or rupture force in sausage processing [9]. Although some studies have improved the gliding properties of natural casings, the structural configuration of the surfaces of natural casings has not been discussed.

Natural casings are by-products of livestock often obtained from a large number of animals from different countries and/or producers. This may lead to the disadvantage of non-uniform strength quality in natural casing products and causes the risk of rupturing during automatic filling in sausage production. However, few previous studies have been performed on toughness variations among natural casings and their influencing factors.

Although studies have been carried out to improve the mechanical properties of natural casings, there is still less research elucidating the fundamental structure–toughness relationship in natural casings. In particular, the reasons for the distinction in toughness has not been clarified from the perspectives of the histological structure, quantitative determination of the components, and comparative discussion among various kinds of natural casings. In this regard, this study investigated the toughness, and biochemical and histological characteristics of hog and sheep casings originated in different production areas (countries) or considering different slaughter ages; the aim was to probe the basic factors influencing the mechanical properties of casings, which might allow the sausage casing quality to be controlled in the natural casing retailer and sausage industry.

## 2. Materials and Methods

### 2.1. Materials

Salted natural hog casings (32–34 mm in diameter; from China, Japan, and USA), sheep casings (20–22 mm; from China, Australia, Egypt, and New Zealand), and lamb casings (20–22 mm; from New Zealand) were used. All materials were obtained from a natural casing retailer, New Asia Trading Co., Ltd., in Japan. The salted casing samples stored at 4 °C were washed and desalted in running water for 1 h at room temperature and then were used for a series of analyses. The sheep and lamb casings used in this study were obtained from sheep older than 2 years and lambs from 1 to 2 years old, respectively.

### 2.2. Toughness Measurement

The desalted casings were cut into one layered sheet; then, the toughness of each casing sheet was measured using a rheometer (NM-2002J; Rheotech) on the middle part. The toughness was indicated by the maximum load value (breaking stress, in N) when a columnar plunger with a diameter of 3 mm was loaded perpendicularly and the plunger broke through the casing. A total of 240 different locations were tested for the toughness evaluation of each sample, which was composed of 12 different animal casings randomly selected from 3 hanks. Each casing for toughness determination was divided by 1 m for histological analyses before being cut into one layered sheet.

### 2.3. Biochemical Analysis

The casings used for measuring toughness were also used to make dried–defatted matter (DDM) for biochemical analyses. The DDM was made as follows: The casings were frozen and crushed with liquid nitrogen; then, the samples were degreased and dried with a mixed solution of chloroform–ethanol (2:1). The biochemical characteristics of the DDM were evaluated (n = 12) by measuring the collagen, elastin, and proteoglycan contents and the heat solubility of collagen according to the methods previously reported by Nishiumi, Kunishima, Nishimura, and Yoshida [10].

#### 2.3.1. Collagen Measurement

According to Hill [11], the powdered DDM was heated in distilled water at 77 °C for 70 min and then separated into insoluble and heat-soluble fractions with two centrifugations. The samples were then hydrolyzed with 6 mol/L HCl at 110 °C for 24 h, and the hydroxyproline content in the hydrolysates was determined [12]. Hydroxyproline values were converted to insoluble and soluble collagen using the coefficients of 7.25 [13] and 7.52 [14], respectively. The total collagen concentration of the samples was determined by adding the amounts of heat-soluble and insoluble collagen. Then, we calculated the percentage of soluble collagen in each sample.

#### 2.3.2. Elastin Determination

The powdered DDM was hydrolyzed in 0.1 mol/L NaOH for 50 min at 98 °C, and the remaining residue was hydrolyzed in 6 mol/L HCI; then, hydroxyproline was quantitated. The concentration of DDM elastin was determined by multiplying the hydroxyproline content by 66.225 [15].

#### 2.3.3. Uronic Acid Content

The DDM was extracted with activated papain (by incubating papain (type III; Sigma) with 0.1 mol/L EDTA) solution for 24 h at 65–70 °C, and the supernatant was collected using centrifugation [16]. The residue washed with 0.1 mol/L phosphate buffer (pH 6.4; containing 0.3 mol/L NaCl) was centrifuged again to collect the supernatant [16]. The uronic acid content in both supernatant fractions was estimated using glucuronolactone as a standard [17].

### 2.4. Histological Analysis

A histological analysis was performed on each sample using the casing left at the time of toughness measurement. As specimens for histochemistry, casings were fixed with 10% formalin-PBS and then stained with Verhoff’s Van Gieson (Elastin Stain Kit; Sigma). Elastin fibers and collagen fibers in each casing were observed under an optical microscope.

Furthermore, to observe the structure of collagen fibers, the casing sample was fixed with a 2% paraformaldehyde–2.5% glutaraldehyde solution (0.1 mol/L phosphate buffer, pH 7.4). According to the cell maceration/ scanning electron microscope (SEM) method of Ohtani, Ushiki, Taguchi, and Kikuta [18], the fixed samples were treated with a 10% NaOH aqueous solution for 5 days and then washed with distilled water for 3 days. The samples were treated with the tannin–osmium method [19] and dehydrated with an alcohol system, and the solution was replaced with t-butyl alcohol for freeze-drying [20]. The dried sample adhered to the sample holder, and gold and palladium were vapor-deposited; then, the sample was observed at an acceleration voltage of 15 kV using a Hitachi S-2380N or Hitachi S-430 scanning electron microscope.

### 2.5. Determination of Pyridinoline Content

The quantification of pyridinoline was performed according to the method of Arakawa, Kim, and Otsuka [21]. Casings were hydrolyzed with hydrochloric acid, and we sufficiently removed the hydrochloric acid using a rotary evaporator. Then, the samples were dissolved in 3 mL of distilled water and filtered through a chromatodisc (0.45 µm pore size). A part of these samples were subjected to hydroxyproline quantification, and the rest were analyzed using a high-performance liquid chromatographic (HPLC) assay. For the HPLC analysis, a Shimadzu LC-10AD HPLC instrument connected with an Inertsil ODS-2 column (4.6 mm I.D. × 25 cm) was used. In the mobile phase, a mixture of 0.1 mol/L sodium phosphate (pH 3.5) and acetonitrile (75:25, *v*/*v*) containing 0.1% SDS and 0.0025% EDTA was used at a flow rate of 1 mL/min. The eluate was monitored with a spectroscopic fluorescence detector (RF-10A; Shimadzu Corporation)—fluorescence excitation at 295 nm and emission at 395 nm. The pyridinoline concentration in the casings was calculated using commercially available pyridinoline (Wako Pure Chemical Industries) as a standard solution and expressed as mole per mole of collagen.

### 2.6. Statistical Analysis

Means and standard errors were calculated in each group. The one-way analysis of variance (ANOVA) was used for statistical analyses using BellCurve for Excel (Social Survey Research Information Co., Ltd.). Significant differences between means were determined with Tukey’s multiple test method at a significance level of *p* < 0.01.

## 3. Results and Discussion

### 3.1. Biochemical and Histological Characteristics of Hog and Sheep Casing Toughness Difference

#### 3.1.1. Hog and Sheep Casing Toughness Properties

To compare and explore the physical properties of natural casings, toughness measurements (breaking stress, in N) were carried out on hog and sheep casings of different origins. As shown in Table 1, the breaking stress was significantly (*p* < 0.01) higher in hog casings than in sheep casings overall. Gzik-Zroska et al. [22] also reported that hog casings have higher breaking stresses than sheep casings because of the individual factors of each species and the various geometry. Feng, Otani, Ogawa, and García-Martín (2021) reported that in their study, the thicknesses of hog and sheep casings were about 0.03 and 0.019 mm, respectively [23]. Moreover, the toughness difference in casings from different animal was reported to be caused by their thicknesses [24]. In both hog and sheep casings from each country, the Chinese ones showed significantly (*p* < 0.01) higher breaking stress. This result was also consistent with the report by Sakata et al. [5] indicating that the Chinese hog casing was harder than the American one. Additionally, the breaking stress of Chinese casings fluctuated greatly, and some of them showed the same value as other domestic casings, but some of them showed even higher breaking stress; as a result, there was a large variation in toughness. On the one hand, there were no significant differences in toughness between hog casings from Japan and the USA and sheep casings from Australia and Egypt.

#### 3.1.2. Biochemical Characteristics of Hog and Sheep Casings and Their Relationship with Toughness

Table 2 shows the total collagen content, the heat-labile collagen content, the heat solubility of collagen, the elastin content, and the uronic acid content of the natural casings.

Both hog and sheep casings (submucosa) contained high contents of collagen (80–92%). No differences in the total collagen content was observed between hog/sheep casings in all samples, even though they had different toughness. A study on total collagen content and distribution in the human colon showed similar results indicating that the percentage mean intensity of total collagen staining in the submucosa was 74 (3.2)% in the adult ascending colon [25].

The elastin content, which mainly constitutes blood vessels, was about 20 mg/g DDM, and this value was as low as the elastin ratio in the intramuscular connective tissue [10]. Since there were almost no differences among all the casing samples, it was considered that elastin is not the major inducing factor of toughness difference among natural casings. In collagen membranes, elastic fibers run parallel to the collagen fibers and the functional interaction of these two fibrous components imparts the tissue with both tensile strength and elasticity [26]. Therefore, elastin may affect toughness in veiny casings that are visibly vascularized.

The uronic acid content, as an index of proteoglycan, did not differ among all casing samples. It has been suggested that certain proteoglycans interact with collagen and contribute to the structural stability of collagen [27]. However, in this study, there were no relationships between the uronic acid content and the toughness variation in the natural casings.

In terms of composition, the natural casing is the connective tissue in the small intestine of livestock and is composed of a large amount of collagen, a small amount of elastin, and proteoglycan. Among all the samples, although there were significant differences in the toughness properties, there were no significant differences in the contents of collagen, elastin, and proteoglycan. From this, we infer that the difference in the toughness property of natural casings has no correlations with the content of its components.

On the other hand, the heat solubility of collagen was significantly (*p* < 0.01) lower in both Chinese hog casings and sheep casings than that in other domestic casings, which corresponded well to the difference in toughness. The heat solubility of collagen indicates what percentage of collagen is solubilized as gelatin by heating at 77 °C for 70 min and is an index indicating the thermal stability of collagen. The thermal stability of collagen is thought to be affected by the formation of collagen intermolecular cross-links and the size and orientation of collagen fibers [28]. Therefore, the heat solubility of collagen is considered to reflect not only the thermal stability of the tissue mainly composed of collagen but also the toughness of the tissue. In other words, it is conjectured that the toughness of the very thin natural casing is caused by the structure of collagen fibers and the formation of intermolecular cross-links.

#### 3.1.3. Histological Characteristics of Hog and Sheep Casings and Their Relationship with the Mechanical Property

The distributions of elastin fibers and collagen fibers in each casing were observed with a light microscope after staining using Elastin Stain Kit. Elastin Stain Kit stains elastin in black and collagen in yellow. It was observed that blood vessels mainly composed of elastin were buried in collagen fibers at low magnification (Figure 1a: green arrow, arteriole; blue arrow, venule). The densities of elastin fibers in arterioles (A) and venules (V) varied (Figure 1c) when dilating the fig of blood vessels. As shown in Figure 1b, a casing is composed of sheets consisting of multiple layers of collagen fibers, where each sheet includes a woven structure of wavy collagen fibers with thin elastic fibers running into it (Figure 1d). The microstructures were similar among all casings (either hog casings and sheep casings or casings from the various production areas). Gunn, Sizeland, Wells, and Haverkamp [24] also reported that sausage casings from different species (cattle, hog, and sheep) are similar in collagen structure. Notably, this characteristic of woven structure with wavy collagen fibers can contribute to the advantages of moisture permeability, smoky ingress, and good elasticity.

Furthermore, a comparative study of the influence of collagen fiber structures in natural casings on different toughness values was performed using a scanning electron microscope. Figure 2 shows the collagen fiber structures of hog casings from Japan and China that showed a significant difference in toughness property. The structure of the collagen fibers on the outside of the casing was similar in the hog casings from both Japan (Figure 2a) and China (Figure 2b) and was composed of a thin sheet layer, in which the collagen fibers ran in parallel. The collagen fibers running on the outside of the hog casings were thicker than those on the inside. On the other hand, the inside of the casing from Japan (Figure 2c) had a spongy-like structure that was formed by collagen fibers aligned in a network, but that was hardly observed in the casing from China (Figure 2d). Collagen fiber distribution on the inner surface of natural casings might affect its slipperiness in the sausage filling processing. A previous study showed that the network-like structure with collagen fibers became visible on the inside surface (mucosal surface) of the porcine small intestinal submucosa and that the inside surface was noticeably smoother than the outside surface (serosal surface) [29].

Figure 3 shows the collagen fiber structure of Australian and Chinese sheep casings that had a significant disparity in terms of toughness. The outside of the Australian sheep casing (Figure 3a) was very finely woven and consisted of tiny collagen fibers. Moreover, in the Chinese sheep casing (Figure 3b), the collagen fibers were relatively thick and had a rough woven shape, which was unlike the orientation of the collagen fibers on the outside of the Australian sheep casing, resembling the appearance of the hog casing. In both Australian and Chinese sheep casings, the inside structure of collagen fibers ran to surround a large hole. However, the Australian sheep intestine (Figure 3c) had a fine, lace-like appearance, while the Chinese sheep intestine (Figure 3d) had thick collagen fibers and a rough appearance. Thimo Maurer et al. reported that the average diameter of collagen fibrils is positively correlated with the tensile strength of the connective tissue [26]. Overall, we surmise that it is the thickness and orientation of collagen fibers that differentiate the mechanical properties of casings. In particular, the Chinese casing was relatively thick with different orientations, thus leading to high toughness and rough appearance.

### 3.2. Toughness and Collagen Characteristics of Sheep and Lamb Casings

#### 3.2.1. Toughness of Sheep and Lamb Casings

As shown in Table 3, the breaking stress of older sheep casings was significantly higher than that of young lamb casings (*p* < 0.01) indicating that casings become tougher with animal age progression. Furthermore, Parry (1988) reported that the general pattern of collagen fibril growth can be established as a function of age [30]. Considering the results of measuring the toughness of national sheep casings, New Zealand lamb, Australian, and Egyptian sheep casings were relatively tender, New Zealand sheep were slightly tougher, and Chinese sheep casings were especially tough. Natural casings are a by-product of livestock, it is not clear at what age they are slaughtered in the natural casing industry. It is believed that in the natural casing industry, animals with different ages show variations in the casing toughness.

#### 3.2.2. Collagen Heat Solubility and Pyridinoline Content in Sheep and Lamb Casings 

As shown in Table 3, there were no significant differences in the total collagen contents in lamb and sheep casings, while the amount of heat-labile collagen content in sheep casings was significantly (*p* < 0.01) lower, and the pyridinoline concentration was significantly (*p* < 0.01) higher than in lamb casings. The differences in pyridinoline content correspond to the toughness disparity between the lamb and sheep casings, indicating that collagen has higher thermal stability in tougher sheep casings. A natural casing is mostly composed of collagen, and the collagen molecules are cross-linked to provide stable strength for the collagen connective tissue. With age progression, the unstable cross-links reduce and are converted to heat-stable mature cross-links (e.g., pyridinoline), which stabilizes the collagen fiber structure and reduces the heat solubility of collagen [31]. Therefore, this study suggests that as the livestock grows up, the casing collagen fiber network becomes stronger, and the casing becomes tougher due to the accumulation of mature cross-links among collagen molecules.

#### 3.2.3. Comparison of Collagen Fiber Structures in Sheep and Lamb Casings

The collagen fiber structures of lamb and sheep casings were observed using a scanning electron microscope (Figure 4). The orientation of collagen fibers on the casing surface was almost the same for lamb (Figure 4a) and sheep (Figure 4b), in that collagen fibers were assembled similar to knitted fabric. However, each collagen fiber in the sheep casing was slightly thicker than in that from lamb. Previous studies reported similar observation on the collagen fiber structure of the submucosa of the small intestine (which is the material of natural casings), where collagen fibers run in a grid pattern similar to woven fabric in all animals [32,33]. In addition, the results support the argument that the lattice orientation of collagen fibers does not change with animal aging, but that the collagen fibers become thicker upon aging [34]. In this study, we also observed the cross-section of lamb and sheep casings. In the cross-sectional scanning electron microscopy study, both the lamb casing (Figure 4c) and the sheep casing (Figure 4d) consisted of several layers of thin collagen fiber sheets. However, sheet layers were large in quantity in the sheep casing, and the thickness of the casing was 0.10 to 0.13 mm for lamb and 0.15 to 0.20 mm for sheep, which was about twice as thick as the lamb casing. This result was also consistent with the report by Feng et al. (2021) indicating that the thickness of sheep casings was about 0.019 mm [23]. These histological features of natural casings are believed to be directly related to the toughness of the casing.

## 4. Conclusions

The present study showed that the main component of hog and sheep casings was collagen with many layers of sheets. The contents of collagen, elastin, and proteoglycan in hog and sheep casings were similar. The variety of the toughness properties of natural casings was due to collagen stability. The toughness of sheep casings was increased by animal age. The collagen stability of sheep casings was enhanced by animal age. Overall, this work shed a light on the structural fundamentals responsible for the toughness of casings, which may benefit the control of casing mechanical properties in the natural casing retailer and sausage industry.

## Figures and Tables

**Figure 1 foods-11-03815-f001:**
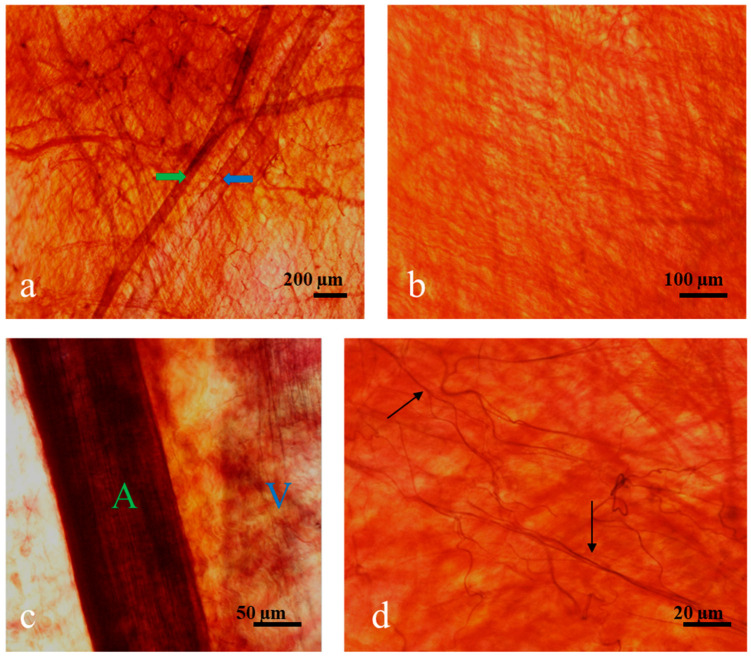
Histochemistry results of connective tissue from natural casings stained with Verhoff‘s Van Gieson stain. Elastin fibers are stained black, and collagen fibers are orange. The photos taken at different magnifications are named a–d, respectively. Arteries, 
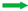
 and A; veins, 
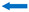
 and V; elastin fibers, 
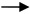
. (**a**) Gross view. Blood vessels, arteries (
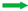
), and veins (
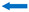
) are embedded in the predominant collagen fibers. (**b**) A casing is mainly composed of numerous layers of sheets of crimped collagen fibers with a crisscross arrangement. (**c**) Elastin fibers in an artery (A) and in a vein (V). (**d**) Fine elastin fibers (
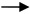
) are scattered sparsely in collagen fibers.

**Figure 2 foods-11-03815-f002:**
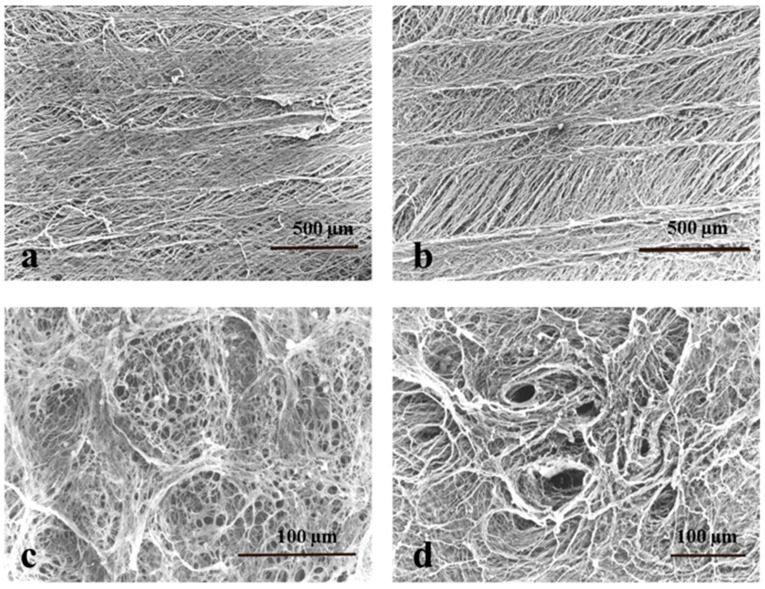
Structures of collagen fibers in Japanese and Chinese hog casings observed using a scanning electron microscope. (**a**,**b**) External surface of hog casings from Japan and China, respectively. (**c**,**d**) Internal surface of hog casings from Japan and China, respectively.

**Figure 3 foods-11-03815-f003:**
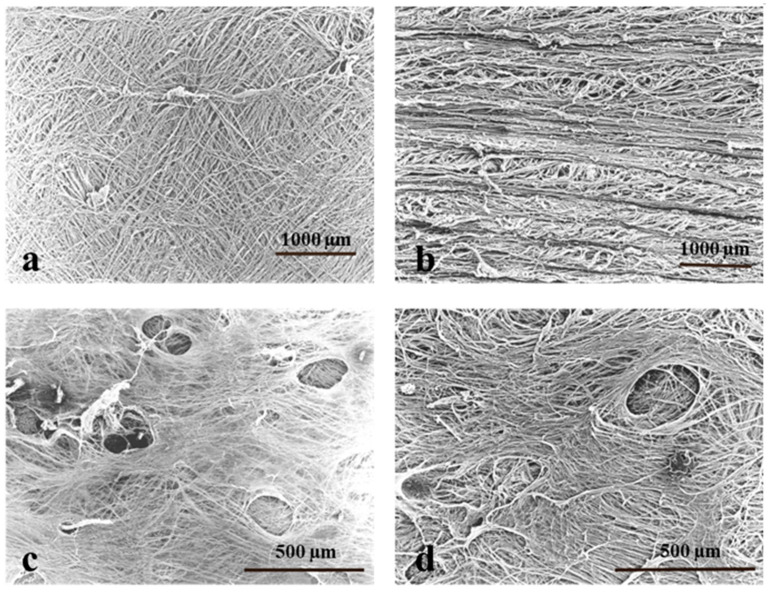
Structures of collagen fibers of Australian and Chinese sheep casings were observed using a scanning electron microscope. (**a**,**b**) External surface of sheep casings from Australia and China, respectively. (**c**,**d**) Internal surface of sheep casings from Australia and China, respectively.

**Figure 4 foods-11-03815-f004:**
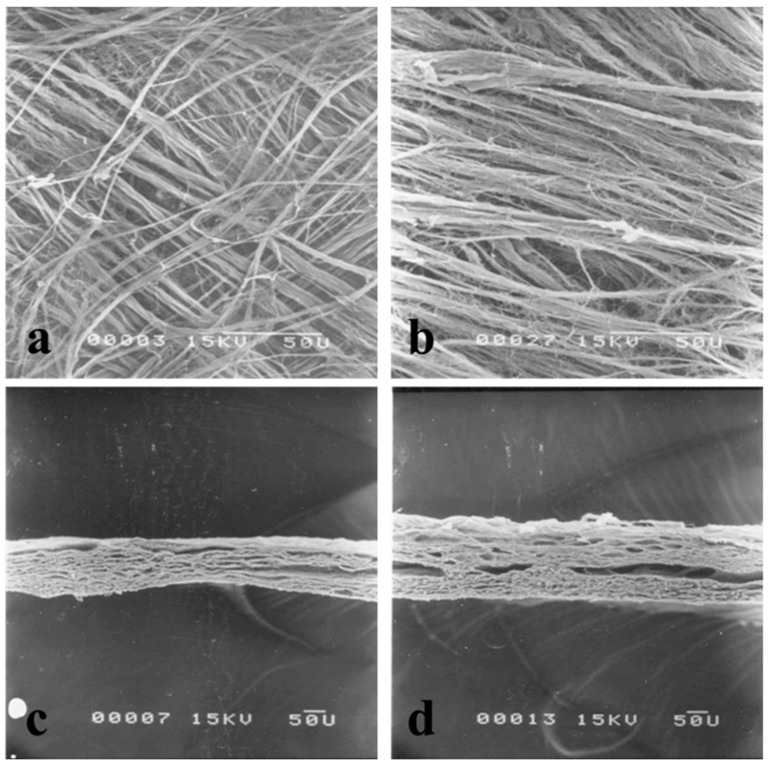
Comparison of collagen fibrous structures of lamb and sheep casings observed using a scanning electron microscope (scale bar, 50 µm). The size and arrangement of collagen fibers on the external surface of lamb and sheep casings are shown in (**a**,**b**), respectively. The thickness of lamb and sheep casings is shown in (**c**,**d**), respectively.

**Table 1 foods-11-03815-t001:** Mechanical characteristics of natural hog and sheep casings.

Origin	Breaking Strength (N)
Hog casing	
China	7.91 ± 1.99 ^A^
USA	6.51 ± 1.06 ^B^
Japan	6.11 ± 1.15 ^B^
Sheep casing	
China	4.89 ± 1.41 ^C^
Egypt	3.23 ± 1.01 ^D^
Australia	3.11 ± 0.74 ^D^

^A–D^: Means with different letters indicate significant differences (*p* < 0.01). Values are presented as means ± standard deviations. n = 12.

**Table 2 foods-11-03815-t002:** Biochemical characteristics of natural hog and sheep casings.

Origin	Total Collagen Content (mg/g DDM)	Heat-Labile Collagen Content (mg/g DDM)	Heat Solubility of Collagen (%, *w*/*w*)	Elastin Content (mg/g DDM)	Uronic Acid Content (mg/g DDM)
Hog casing					
China	921 ± 42 ^A^	14.4 ± 5.0 ^A^	1.56 ± 0.26 ^A^	22.0 ± 1.6 ^A^	1.72 ± 0.34 ^A^
USA	830 ± 101 ^A,B^	19.1 ± 6.3 ^A^	2.30 ± 0.45 ^B^	N.D.	1.82 ± 0.24 ^A^
Japan	837 ± 82 ^A,B^	18.5 ± 4.4 ^A^	2.12 ± 0.36 ^B^	16.3 ± 0.7 ^B^	1.99 ± 0.27 ^A^
Sheep casing					
China	868 ± 92 ^A,B^	19.7 ± 8.1 ^A^	2.27 ± 0.61 ^B^	23.9 ± 1.2 ^A^	1.86 ± 0.15 ^A^
Egypt	844 ± 86 ^A,B^	30.1 ± 6.5 ^B^	3.57 ± 0.58 ^C^	N.D.	1.72 ± 0.17 ^A^
Australia	803 ± 64 ^B^	29.0 ± 6.2 ^B^	3.61 ± 0.37 ^C^	22.8 ± 2.1 ^A^	1.79 ± 0.37 ^A^

^A–C^: Means with different letters indicate significant differences (*p* < 0.01). N.D.: not determined. Values are presented as means ± standard deviations. n = 12.

**Table 3 foods-11-03815-t003:** Toughness property and collagen characteristics of natural lamb and sheep casings.

New Zealand Casing	Breaking Strength (N)	Total Collagen Content (mg/g DDM)	Heat-Labile Collagen Content (mg/g DDM)	Heat Solubility of Collagen (%, *w*/*w*)	Pyridinoline Concentration (Mole/Mole of Collagen)
Lamb	2.93 ± 0.83 ^A^	865 ± 27 ^A^	94.4 ± 7.8 ^A^	10.91 ± 0.42 ^A^	0.046 ± 0.003 ^A^
Sheep	3.99 ± 1.18 ^B^	884 ± 51 ^A^	28.4 ± 4.0 ^B^	3.32 ± 0.22 ^B^	0.128 ± 0.009 ^B^

^A,B^: Means with different letters indicate significant differences (*p* < 0.01). Lamb, 1–2 years old; sheep, older than 2 years. Values are presented as means ± standard deviations. n = 12.

## Data Availability

Data is contained within the article.

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
