# Peer review of "Toughness Variations among Natural Casings: An Exploration on Their Biochemical and Histological Characteristics"

_foods, 2022, doi:10.3390/foods11233815_

Round 1

Reviewer 1 Report

This is a work with an interest in understanding the structure of different types of casings, with a large amount of laboratory work carried out and interesting results especially for the casing retailer and its customers.

Although the objective is to probe the basic factors that influence the mechanical properties of the casings, from the experimental design and the results it is not possible to see how the precise quality control of casings for sausages in the meat industry can be achieved.

The introduction with a review of the state of the art is focused on the relationship between the different types of casings used with the physical and sensory quality of the final product, supported by old bibliographic references and that has little relationship with the objective of the study.

The type of sampling methods should be described in terms of sampling time, storage conditions of the material at the retailer, number of samples, number of repetitions and number of repetitions, etc.

The model of statistical analysis should be described.

Basically, the results are the comparison of the hardness and chemical characteristics of different sausage casings. At no time can you see any possibility of "precise control" of the quality of sausage casings in the meat industry. Anyway, conclusions should not be a discussion of the results and the suggestion that the increased thermal and structural stabilities of collagen with animal growth contribute to the toughness properties of casings is not valid as a result of the experimental research.

I believe that under the conditions in which the research objectives were stated, it is only a relative interest, since it does not bring any new methodology, nor any new knowledge, being only a biochemical and histological characterization of different sausage casings.

Author Response

请参阅附件。

Reviewer 2 Report

The article is well written and discusses the differences found among some mechanical properties on casings.

The information reported is clear to me. Just a few things related to language that should be revised:

Line 67. The abbreviation "etc." maybe is not necessary

Line 189. The comparative "higher" maybe does not fit the sentence

Lines 245 and 258. The superlative "(toughest)" maybe does not fit the sentence

Author Response

Cover letter

14 October 2022

Dear reviewer

Thank you for your kind suggestions and comments.

The suggestions you mentioned have been completely revised point by point.

  • Line 67: "etc." has been delated.
  • Line 189: "higher" has been changed to "high".
  • Lines 245 and 258: "toughest" has been delated.

 The revised details are displayed in the manuscript (Please see the attachment) by the "Track Changes" and "Displaying comments" functions of MS Word.

If you have any suggestions or comments, please do let me to know. Thank you very much.

Yours sincerely,

Tadayuki Nishiumi Ph.D.

Graduate School of Science and Technology, Niigata University, 8050, Ikarashi 2-no-cho Nishi-ku, Niigata City 950-2181, Japan

Tel: +812-5262-6663;

E-mail: riesan@agr.niigata-u.ac.jp

Reviewer 3 Report

Comments to the Authors

Title 

The Toughness Varied Among Natural Casings: An Exploration on Their Biochemical, Histological Characteristics

The manuscript submitted by Liu et al. investigated the toughness, biochemical, and histological characteristics of hog and sheep casings from different production areas or from different ages. Although the paper was well written and logistically organized, some errors and problems need to be rectified.

Line 10. Please include the objective at the beginning of the abstract

Line 13. (p < 0.01)

Line 14. (p < 0.01)

Line 17. (p < 0.01)

Line 19. (p < 0.01)

Lines 91. I suggest rewriting this section to avoid a high percentage of plagiarism. This section is almost like Nishiumi et al. (1995). Intramuscular connective tissue components contributing to raw meat toughness in various porcine muscles. Animal Science and Technology, 66(4), 341-348. 

Line 161. (p < 0.01)

Line 168. (p < 0.01)

Line 172. (p < 0.01)

Line 173-174. Could the authors explain the reason?

Lines 182. (p < 0.01)

Line 182. Include: “Values are presented as mean ± standard deviation” (or standard error)

Lines 196-197. Support this idea with a reference

Line 203. (p < 0.01)

Line 207. Minutes should be min

Line 215. (p < 0.01)

Line 215. Include: “Values are presented as mean ± standard deviation” (or standard error)

Lines 222, 224, 225, 227. Fig. should be Figure 

Lines 236, 238-241. Fig. should be Figure 

Lines 24-249, 251. Fig. should be Figure

Lines 260, 261, 265, 266. Fig. should be Figure

Line 278. (p < 0.01)

Line 287. (p < 0.01)

Line 288. Include: “Values are presented as mean ± standard deviation” (or standard error)

Line 293, 294. (p < 0.01)

Line 305. (p < 0.01)

Line 305. Include: “Values are presented as mean ± standard deviation” (or standard error)

Lines 309, 310, 319. Fig. should be Figure

Lines 329, 329. Fig. should be Figure

Lines 331-355. The conclusion is not a summary of the results. Rewrite the conclusion section

Line 336. Update all references

Author Response

Cover letter

14 October 2022

Dear reviewer

Thank you for your kind suggestions and comments.

The suggestions you mentioned have been completely revised point by point.

  • Line 10: The objective has been added at the beginning of the abstract.
  • "P < 0.01" has been changed to "p < 0.01", throughout the whole manuscript.
  • Line 91: The section of "2.3 Biochemical Analysis" has been rewritten to avoid a high percentage of plagiarism.
  • Line173-174: There was no report to clarify the characteristics of differences in toughness between casings of the same animal origin.
  • “Values are presented as mean ± standard deviation” has been added throughout the whole tables.
  • Line 196-197: we added a reference to support this idea.
  • Line 207: "minutes" has been changed to "min".
  • "Fig" has been changed to "Figure", throughout the whole manuscript.
  • Lines 331-355: The conclusion has been rewritten.
  • Line 336: All references have been updated.

 The revised details are displayed in the manuscript (Please see the attachment) by the "Track Changes" and "Displaying comments" functions of MS Word.

If you have any suggestions or comments, please do let me to know. Thank you very much.

Yours sincerely,

Tadayuki Nishiumi Ph.D.

Graduate School of Science and Technology, Niigata University, 8050, Ikarashi 2-no-cho Nishi-ku, Niigata City 950-2181, Japan

Tel: +812-5262-6663;

E-mail: riesan@agr.niigata-u.ac.jp

Reviewer 4 Report

Dear authors, Please find the attached file.

Author Response

Cover letter

14 October 2022

Dear reviewer

Thank you for your kind suggestions and comments.

The suggestions you mentioned have been completely revised point by point.

  • Line 10: What is the aim of the study?

→ The aim of the study has been added.

  • Line 14: P italic throughout the whole manuscript.

→ Yes. P italic throughout the whole manuscript.

  • Line 15: Is this result?

→ No, this sentence has been delated (The abstract has been rewritten).

  • Line 23: Add comma before which all over the whole manuscript.

→ Comma has been added before “which” all over the whole manuscript.

  • Line 75: Is this diameter having any effects on the results?

→ Generally, the toughness of casing is influenced by their diameters.

  • Line 75: Delete the

→ “the” has been delated.

  • Line 78: What are these materials?

→ "All materials" means the natural casing samples above-mentioned.

  • Line 79: Remove casings here to avoid repetition.

→ "casings" has been removed.

  • Line 82: Where is the reference for this method?

→There is no reference. We designed the method.

  • Line 84: Remove by.

→ "by" has been removed.

  • Line 169: Add more references for discussion.

→ More references have been added for discussion.

  • Line 184: The authors must provide more studies for discussion.

→ More references have been added for discussion.

  • Line 244: Where is the reference for the comparative study?

→ Reference has been added.

  • Line 258: Where is the discussion for this result?

→ Reference has been added.

  • Line 276: Where is the discussion for this result?
  • → Reference has been added.
  • Line 286: Data is simple to be illustrated in table.

→ The table 3 and table 4 have been combined to Table 3.

  • Lie 330: Brief only the most vital findings of the study. Not all results.

→ The conclusion has been rewritten.

  • Line 355: Where is the ethical approval?

→ The ethical approval has been added.

  • Line 366: Add more recent references.

→ More recent references have been added.

 The revised details are displayed in the manuscript (Please see the attachment) by the "Track Changes" and "Displaying comments" functions of MS Word.

If you have any suggestions or comments, please do let me to know. Thank you very much.

Yours sincerely,

Tadayuki Nishiumi Ph.D.

Graduate School of Science and Technology, Niigata University, 8050, Ikarashi 2-no-cho Nishi-ku, Niigata City 950-2181, Japan

Tel: +812-5262-6663;

E-mail: riesan@agr.niigata-u.ac.jp

Round 2

Reviewer 1 Report

I apologize, but the author's answers are not enlightening enough to change my opinion. I continue to consider that the manuscript is of only relative interest, as it does not bring any new methodology, nor any new knowledge, being only a biochemical and histological characterization of different sausage casings.

Reviewer 4 Report

Dear authors, 

Thank you for considering my comments. Please, reconsider the conclusion again without mentioning the aim and steps of the research. 
